# Network-wide reorganization of procedural memory during NREM sleep revealed by fMRI

**Shahabeddin Vahdat[1], Stuart Fogel[2], Habib Benali[3,4], Julien Doyon[1]***

[1]Functional Neuroimaging Unit, Cenre de recherche, Institut universitaire de gériatrie de Montréal , Université de Montreal, Québec, Canada; [2]School of Psychology, University of Ottawa, Ontario, Canada; [3]PERFORM Centre, University of Concordia, Montreal, Quebec, Canada; [4]INSERM/UPMC, Pitié-Salpêtrière Hospital, Paris, France

**Abstract** Sleep is necessary for the optimal consolidation of newly acquired procedural memories. However, the mechanisms by which motor memory traces develop during sleep remain controversial in humans, as this process has been mainly investigated indirectly by comparing pre- and post-sleep conditions. Here, we used functional magnetic resonance imaging and electroencephalography during sleep following motor sequence learning to investigate how newly-formed memory traces evolve dynamically over time. We provide direct evidence for transient reactivation followed by downscaling of functional connectivity in a cortically-dominant pattern formed during learning, as well as gradual reorganization of this representation toward a subcortically-dominant consolidated trace during non-rapid eye movement (NREM) sleep. Importantly, the putamen functional connectivity within the consolidated network during NREM sleep was related to overnight behavioral gains. Our results demonstrate that NREM sleep is necessary for two complementary processes: the *restoration* and *reorganization* of newly-learned information during sleep, which underlie human motor memory consolidation.

**\*For correspondence:** julien. doyon@umontreal.ca

## Introduction

There is now ample evidence that sleep plays a crucial role in the consolidation of newly-acquired procedural memory, particularly for explicitly instructed sequential motor skills (*Walker et al., 2002*; *Korman et al., 2003*; *Doyon and Benali, 2005*; *Korman et al., 2007*; *Debas et al., 2010*). Several mechanistic hypotheses have also been proposed regarding the contribution of sleep in this memory process (see [*Frankland and Bontempi, 2005*; *Rasch and Born, 2007*; *Tononi and Cirelli, 2014*] for comprehensive reviews). Yet, the dynamic neural changes that drive motor memory consolidation during sleep still remain controversial (*Frankland and Bontempi, 2005*; *Rasch and Born, 2013*; *Tononi and Cirelli, 2014*).

One pioneering sleep-dependent consolidation model, the *trace reactivation hypothesis* assumes that the repeated reactivation of a recently formed memory representation during sleep leads to a gradual strengthening of the learning-related connections, and thus to long-term storage of the memory trace (*Rasch and Born, 2007*, *2013*). There is now mounting evidence in support of this hypothesis including the replay of hippocampal place cell firing (*Skaggs and McNaughton, 1996*; *Lee and Wilson, 2002*) in rodents, as well as human studies employing targeted memory reactivation paradigms using auditory or olfactory cues (*Rasch et al., 2007*; *Cousins et al., 2014*; *Laventure et al., 2016*), and neuroimaging studies showing the reactivation of learning-related brain

**eLife digest** The idea that, while you sleep, you could be honing skills such as the ability to play a musical instrument may sound like science fiction. But studies have shown that sleep, in addition to being beneficial for physical and mental health, also enhances memories laid down during the day. The process by which the brain strengthens these memories is called consolidation, but exactly how this process works is unclear.

Memories are thought to persist as altered connections between neurons, often referred to as memory traces. When we practice a skill, we activate the neurons encoding that skill over and over again, strengthening the connections between them. However, if this process were to continue unchecked, eventually the connections would become saturated and no further increases in strength could occur. One possible solution to this problem is that sleep enhances skill learning by downscaling connections across the brain as a whole, thereby freeing up capacity for further learning. Alternatively, sleep may reorganize an initially unstable memory trace into a more robust form with the potential to last a lifetime.

To test these possibilities, Vahdat et al. asked healthy volunteers to practice a finger-tapping task while lying inside a brain scanner, and then to sleep inside that scanner for 2–3 hours. When the volunteers returned to the scanner the next morning and attempted the task again, they performed better than they had the previous night. Their brains also showed a different pattern of activity when performing the task after a night's sleep.

So what had happened overnight? As the volunteers lay awake inside the scanner, their brains reactivated the memory trace formed during learning. However, as they entered a stage of non-dreaming sleep called non-REM sleep, this activity became weaker. At the same time, a new pattern of activity – the one that would dominate the scan the next morning – began to emerge. Whereas the post-learning activity was mainly in the brain's outer layer, the cortex, the new pattern included other areas that are deeper within the brain. The activity of one deeper region in particular, the putamen, predicted how well the volunteers would perform the task the next day.

Non-REM sleep thus strengthens memories via two complementary processes. It suppresses the initial memory trace formed during learning, and reorganizes the newly-learned information into a more stable state. These results might explain why people who are sleep-deprived often have impaired motor skills and memories. The findings also open up the possibility of enhancing newly learned skills by manipulating brain circuits during non-REM sleep.

regions during sleep or awake rest (*Maquet et al., 2000*; *Rasch et al., 2007*; *Deuker et al., 2013*; *Staresina et al., 2013*; *Tambini and Davachi, 2013*).

Another model, built in part upon the trace reactivation, the *systems consolidation hypothesis* (*Frankland and Bontempi, 2005*; *et al., 2005*; *Rasch and Born, 2013*) proposes that sleep engages an active reorganization process that stabilizes the labile neural representation of a novel skill into a consolidated memory trace. For instance, a systematic transfer in memory representations from hippocampal to neocortical areas has been reported for non-procedural forms of memories (*Frankland et al., 2004*; *Maviel et al., 2004*; *Frankland and Bontempi, 2005*). On the other hand, a systemic shift from cortical (e.g., motor, parietal cortex) to subcortical regions (e.g., striatum) has been proposed to underlie the consolidation of procedural memory, and motor sequence learning in particular (*Doyon and Benali, 2005*; *Yin et al., 2009*; *Debas et al., 2010*; *Kawai et al., 2015*). Yet in humans, the systems consolidation model has only been inferred indirectly by comparing the effect of motor practice on offline gains in behavioral performance and changes in neural activity between the initial learning and retention sessions separated by either diurnal or nocturnal sleep (*Walker et al., 2002*; *Fischer et al., 2005*; *Gais et al., 2007*; *Takashima et al., 2009*; *Debas et al., 2010*). Thus, direct evidence in support of this hypothesis from human neuroimaging studies is lacking.

Finally, an alternative and potentially complementary model, *the synaptic homeostasis hypothesis* (*Tononi and Cirelli, 2003*, *2006*, *2014*) proposes that local neuronal networks are potentiated and eventually become saturated during learning. In order for new information to be encoded the

following day, sleep would be involved in the *restoration* of these local networks by downscaling the strength of synaptic connections (*Tononi and Cirelli, 2003*; *Huber et al., 2004*; *Tononi and Cirelli, 2006*). However, direct experimental evidence to support the synaptic homeostasis hypothesis in humans remains limited and controversial (*Frank, 2012*). It is thus unclear whether and how these different sleep-dependent mechanisms of memory consolidation may be reconciled and contribute to motor skill learning in humans. Here, for the first time, we used simultaneous EEG and fMRI in order to identify the relative contributions of the trace reactivation, systems consolidation, and synaptic homeostasis hypotheses to the consolidation of procedural memory in humans. Specifically, we tested the hypothesis that the memory trace of motor sequence learning involves network-wide reactivation and further reorganization into a more stable representation during non-rapid eye movement (NREM) sleep periods.

## Results

### Methods: overall experimental approach

In order to directly examine the off-line periods during which the motor memory trace is being consolidated, we acquired blood-oxygen-level dependent (BOLD) fMRI data during motor sequence task practice, wake resting-state and post-training sleep conditions. Brain functional images were recorded while thirteen participants performed two different finger movement tasks using a response pad one week apart. In the motor sequence learning (MSL) task, subjects practiced a self-paced, explicitly known 5-item finger sequence task, which was compared with performance on a motor control task (CTL) in which participants were asked to produce simultaneous movements of all four fingers at the same average frequency, and for the same number of times as in the MSL task. These two conditions were administered in a counterbalanced order (*Figure 1a*). For both MSL and CTL tasks, participants underwent an initial training session at 10:30 PM (i.e., learning session; S1), followed by a retest session at 9:00 AM the next morning (i.e., retest session; S2) (*Figure 1a*). Resting-state conditions, during which subjects stayed awake with eyes opened, were also acquired before and after each practice session in the evening (RS1 and RS2) and the following morning (RS3 and RS4; *Figure 1a*). Immediately following the training session (i.e., around 11:00 PM), a simultaneous EEG-fMRI recording scan lasting a maximum of 2.5 hr took place while subjects slept in the scanner. This design allowed us to investigate MSL memory trace reactivation and further transformation during off-line periods, including both resting-state and sleep conditions, from its initial state to a consolidated trace that was later recruited during performance at retest.

### Behavioral results

Motor performance was measured using the speed at which key presses for correct responses were executed in both MSL and CTL tasks (*Albouy et al., 2012*; *Fogel et al., 2014*; *Lungu et al., 2014*). As expected, a two-factor repeated measures ANOVA across practice blocks (14 blocks) and task conditions (MSL, CTL) during the training session (S1) revealed that performance speed evolved differently over the course of learning between the two tasks (*Figure 1b*; significant condition $\times$ practice block interaction; $F_{13,156} = 4.94$, $p < 0.0001$). Yet given that we intentionally matched the speed at which both MSL and CTL tasks had to be performed, as expected, average speed did not differ between tasks (no main effect of task; $F_{1,12} = 3.14$, $p > 0.1$). Also, consistent with previous studies (*Walker et al., 2002*; *Albouy et al., 2015*), only performance on the motor learning task revealed evidence of consolidation overnight, as indicated by off-line improvements in the MSL, but not the CTL task, in the absence of additional practice (*Figure 1b*). Specifically, the improvement in task performance between the end of the training (mean of last three blocks in S1) and beginning of the retest session (mean of first three blocks in S2) significantly differed across tasks, as revealed by a two-factors [session (end S1, beginning S2) $\times$ task (MSL, CTL)] repeated measures ANOVA (significant interaction; $F_{1,12} = 16.77$, $p = 0.001$; follow-up paired t-tests: MSL[beginning S2 – end S1], $t_{12} = 2.43$, $p < 0.05$; and CTL[beginning S2 – end S1], $t_{12} = -1.65$, $p > 0.1$; *Figure 1c*).

We also examined changes in performance accuracy by measuring the percentage of incorrect key presses in each block of the MSL task. Given that the simple 5-item sequence was explicitly known to the participants, as expected (*Walker et al., 2002*; *Debas et al., 2010*; *Albouy et al., 2012*; *Fogel et al., 2014*), performance error was very low overall, and did not show any significant

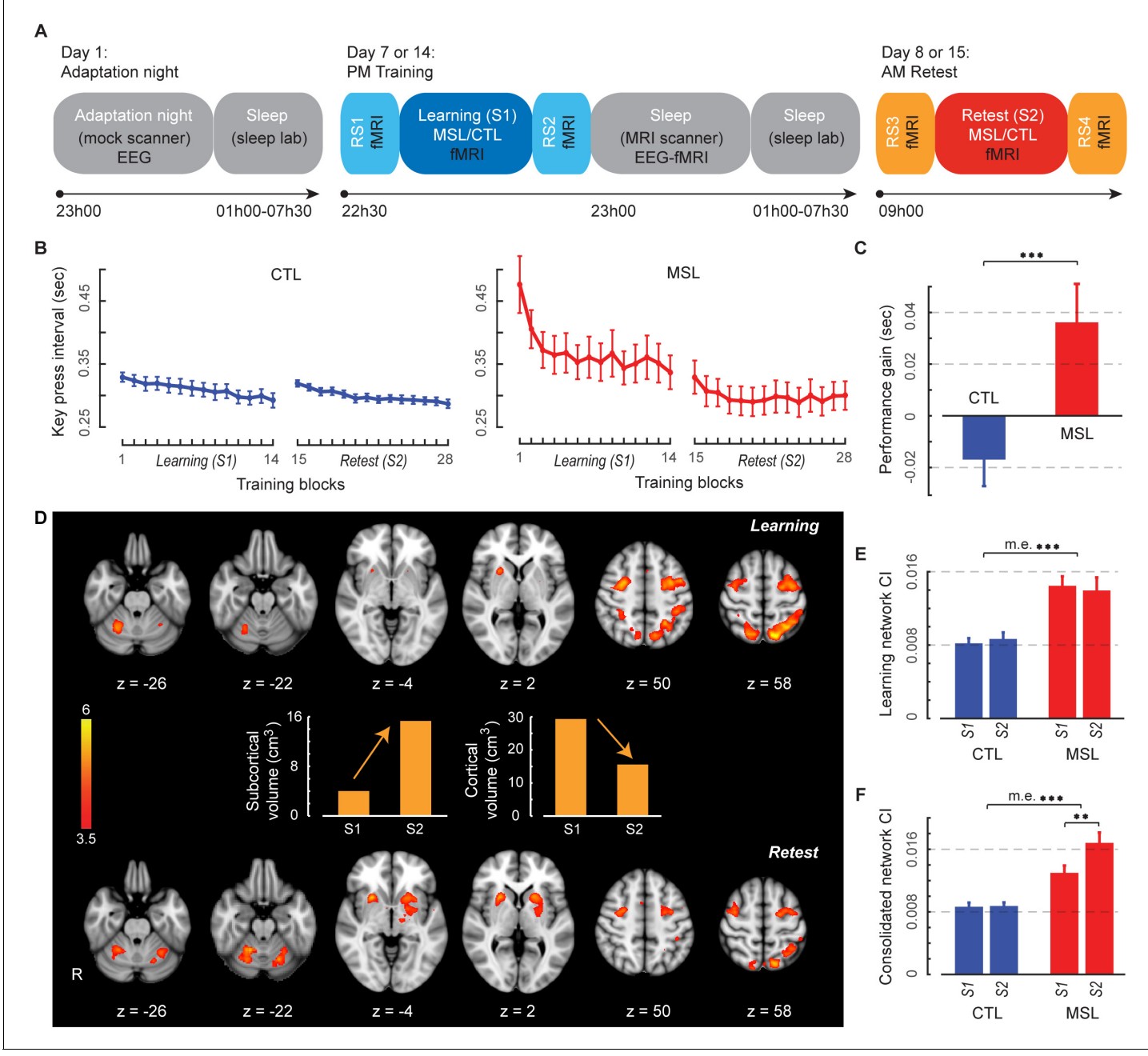

**Figure 1.** Experimental design, behavioral performance, and task activation maps. (**A**) Experimental procedure. On Day 1, subjects first experienced a screening and adaptation night in the mock scanner, which mimicked conditions experienced in both experimental and control nights. Subjects returned and underwent fMRI scans (Day seven and Day 14) while training (S1) on either the motor skill learning (MSL) or motor control (CTL) task in a counterbalanced order, interleaved by resting-state conditions (RS1 and RS2). This was followed by simultaneous EEG-fMRI sleep recording for up to ~2.5 hr. Subjects were then allowed to sleep for the remainder of the night in the sleep lab. Finally, on the following morning subjects underwent retest fMRI sessions (S2) on the same take as the previous training session (Day eight and Day 15), interleaved by resting-state conditions (RS3 and RS4). Arrows shows the experiment's timeline. (**B**) Performance speeds (i.e., inter-key interval) averaged across all subjects show that the learning curves differed between the MSL (red) and CTL (blue) conditions during the learning session (S1). (**C**) Only the MSL task was consolidated overnight, as indicated by performance gain averaged across subjects (asymptotic performance at the end of S1 compared to the beginning of S2). (**D**) Color-coded activation maps representing motor sequence-related areas during the learning (S1) and retest (S2) sessions (corrected for multiple comparisons using Gaussian random field theory, cluster level threshold $p<0.05$). Bar plots illustrate the volume of cortical and subcortical activation in each map. As expected, the connectivity index (CI) within the learning (**E**) and the consolidated (**F**) patterns was significantly higher in the MSL compared to the CTL condition. Error bars represent s.e.m.; ** and *** indicate $p<0.01$ and $p<0.001$, respectively.

*Figure 1 continued*

The following source data and figure supplements are available for figure 1:

**Source data 1.** Summary of activation peaks related to the learning pattern.
**Source data 2.** Summary of activation peaks related to the consolidated pattern.
**Figure supplement 1.** Task-related activation maps during the learning session (S1).
**Figure supplement 2.** Differences in the activation of motor sequence-related areas between the learning (S1) and retest (S2) sessions.

change overnight (average error in the last three blocks of S1 compared to the first three blocks of S2, $t_{12} = 0.95$, p>0.35). This further confirms that the performance speed was a suitable measure to quantify off-line improvements in motor performance. Furthermore, we investigated performance variability by calculating the standard deviation of inter-key-press intervals in each block of MSL and CTL tasks. In line with the performance speed results, performance variability significantly decreased overnight only in the MSL task, as revealed by a two-factors [session (end S1, beginning S2) × task (MSL, CTL)] repeated measures ANOVA (significant interaction; $F_{1,12} = 15.9$, p = 0.0018; follow-up paired t-tests: MSL [beginning S2 – end S1], $t_{12} = 3.98$, p<0.002; and CTL [beginning S2 – end S1], $t_{12} = 0.36$, p>0.7).

## Learning and consolidated activation patterns

We identified distinct brain activation patterns recruited during the learning and retest sessions following a night of sleep. For comparison purpose, the task-related activation maps during MSL and CTL sessions are presented in *Figure 1—figure supplement 1*. Using a two-factor [practice session (S1, S2) × task (MSL, CTL)] repeated-measures ANOVA at the group level, we identified motor sequence-related brain areas that were either activated during the learning (the '*learning pattern*'; S1[MSL – CTL], *Figure 1d* top) or retest session (the '*consolidated pattern*'; S2[MSL - CTL], *Figure 1d* bottom). Although, both the learning and consolidated patterns comprised similar sensorimotor core regions (see *Figure 1—source data 1* and *2*), the relative activation levels of different cortical and subcortical areas were mostly altered across the two maps; that is, the consolidated pattern revealed increased activity in sub-cortical structures and decreased activity in cortical regions (*Figure 1d*). Specifically, despite the fact that the total volume of motor sequence-related activity was preserved across sessions (37.48 cm$^3$ in learning versus 36.10 cm$^3$ in consolidated pattern), the volume of all cortically activated voxels (including mostly the fronto-parietal sensorimotor regions) in the consolidated pattern was almost reduced by half (from 29.34 cm$^3$ to 15.58 cm$^3$), while the volume of sub-cortical activations (including mostly the basal ganglia and cerebellar regions) was nearly quadrupled (from 4.01 cm$^3$ to 15.32 cm$^3$; *Figure 1d* middle bar plots). Similarly, a contrast between the learning and retest sessions strongly confirmed our volume-based analysis results; that is, two cortical clusters (including the superior parietal lobule and anterior intraparietal sulcus bilaterally) showed significantly greater activation during the learning compared to the retest session, while only subcortical regions (including the putamen and cerebellar cortex) revealed significantly greater activation during the retest compared to the learning session (*Figure 1—figure supplement 2*). Consistent with recent work in both human and animal models (*Debas et al., 2010*; *Kawai et al., 2015*), these results suggest a topological shift of activity from cortical to subcortical regions that might underlie sequence memory consolidation.

To investigate whether regions within these distinct patterns became more highly interconnected (reflecting a strengthening of the memory trace), we examined the changes in functional connectivity within the learning and the consolidated patterns in both MSL and CTL conditions. The strength of functional connectivity within a given brain network (i.e., the 'connectivity index', (CI)) was estimated using a straightforward approach that measured the overall co-activation level of brain areas within that network during different fMRI runs (*Vahdat et al., 2014*). Specifically, the CI was defined as the power of a time series of normalized coefficients in a spatial regression model, which estimated the co-activation level of areas within a given network over different scanning times (see Materials and methods for the formulation). This connectivity measure was selected as it provides a

hypothesis-driven multivariate approach specifically suited to study dynamics of changes in connectivity within a network of areas across the whole brain (see Materials and methods for more details).

As a validation check, we first evaluated CI during task performance in the learning (S1) and retest (S2) sessions. It was expected that since the learning and consolidated patterns are extracted from the [MSL – CTL] contrast, we would observe greater levels of CI during the MSL as compared to the CTL task periods. Consistently, a two-factors (session × task) repeated measures ANOVA reported a significant main effect of task for both the learning ($F_{1,12} = 31.2$, p<0.0005; *Figure 1e*) and consolidated patterns ($F_{1,12} = 63.4$, p<0.000005; *Figure 1f*). Notably, there was also a significant effect of session for the consolidated pattern, showing greater CI during the retest (S2) compared to the learning (S1) session (significant interaction; $F_{1,12} = 9.33$, p = 0.01; also significant main effect of session in MSL, $t_{12} = 3.31$, p = 0.006). This analysis confirms that CI is a sensitive measure to detect changes in within-network functional connectivity across experimental conditions.

## Sleep-dependent reactivation and reorganization of the memory trace

In order to investigate whether memory reorganization from the learning to the consolidated trace occurred during the off-line periods (dependent upon either simple passage of time or sleep), or whether the consolidated trace merely manifested itself during retest, we calculated CI within the learning and consolidated patterns during different resting-state periods (RS1, RS2, RS3), as well as during NREM sleep. A two-factor (resting-state condition (RS1, RS2, RS3) × task (MSL, CTL)) repeated-measures ANOVA revealed that the CI changed as a function of motor task condition (MSL vs. CTL) across resting-state periods for the consolidated pattern (significant interaction; $F_{2,24} = 6.06$, p = 0.007, *Figure 2b*). Interestingly, CI within the consolidated pattern was significantly enhanced for the MSL task only during RS3 (significant MSL [RS3-RS1], $t_{12} = 3.14$, p<0.01, significant MSL [RS3-RS2], $t_{12} = 2.26$, p<0.05, and significant main effect of task during RS3, $t_{12} = 3.78$, p<0.005, *Figure 2b*). Thus, despite the fact that the consolidated pattern's CI was not yet increased immediately after training (MSL [RS2-RS1], $t_{12} = 0.94$, p>0.35), it was already significantly elevated *before* the retest session (i.e. RS3), hence suggesting that the consolidation process took place during the preceding interval filled with sleep, and did not manifest itself as a result of practice during retest.

By contrast, the CI analysis within the learning pattern yielded an opposite pattern of findings. Although the effect of motor task condition (MSL vs. CTL) across all resting-state conditions was only marginally significant (interaction; $F_{2,24} = 3.14$, p = 0.06, *Figure 2a*), there was a significant effect of task on the learning pattern's CI across the resting-state conditions before and after learning (repeated measured ANOVA (resting-state condition (RS1, RS2) × task (MSL, CTL); $F_{1,12} = 6.99$, p = 0.02, *Figure 2a*). A follow-up paired *t*-test revealed that CI increased immediately following learning in the MSL task only (significant MSL [RS2-RS1], $t_{12} = 3.05$, p = 0.01, and significant RS2 [MSL - CTL], $t_{12} = 2.37$, p = 0.035). However, the learning pattern's CI dropped after sleep, so that it was no longer significantly different from baseline in the following morning (MSL [RS3-RS1], $t_{12} = 1.6$, p>0.1), nor was it different across MSL and CTL tasks in the post-sleep resting-state condition (RS3 [MSL - CTL], $t_{12} = 1.1$, p>0.25).

The resting-state analyses suggest that the sequence-related memory trace was likely reorganized from the learning to the consolidated pattern between RS2 and RS3 runs, that is, during sleep. To directly test this hypothesis, we calculated CI during stage 2 and slow wave sleep (SWS) of NREM sleep, as classified by simultaneous EEG recordings during the intervening sleep session. Importantly, this analysis confirmed that only the consolidated pattern's CI was significantly elevated during NREM sleep in the MSL compared to the CTL night ($t_{12} = 3.47$, p<0.005; *Figure 2d*). By contrast, the learning pattern's CI during NREM sleep did not differ significantly between the two tasks ($t_{12} = 1.26$, p>0.2; *Figure 2c*).

In order to test the specificity of our findings with respect to other activation patterns that would be presumably unrelated to the experimental conditions, we extracted four highly-reproduced canonical brain networks (*Damoiseaux et al., 2006*) using the application of independent component analysis (ICA) on the task fMRI data (see Materials and methods). These networks included the default mode, visual, and the left and right fronto-parietal networks (*Figure 2—figure supplement 1*). We then calculated CI within each of these networks during different resting-state periods (RS1, RS2, RS3), as well as during NREM sleep. Two-factor (resting-state condition (RS1, RS2, RS3) × task (MSL, CTL)) repeated-measures ANOVAs revealed no significant change in CI in any of these

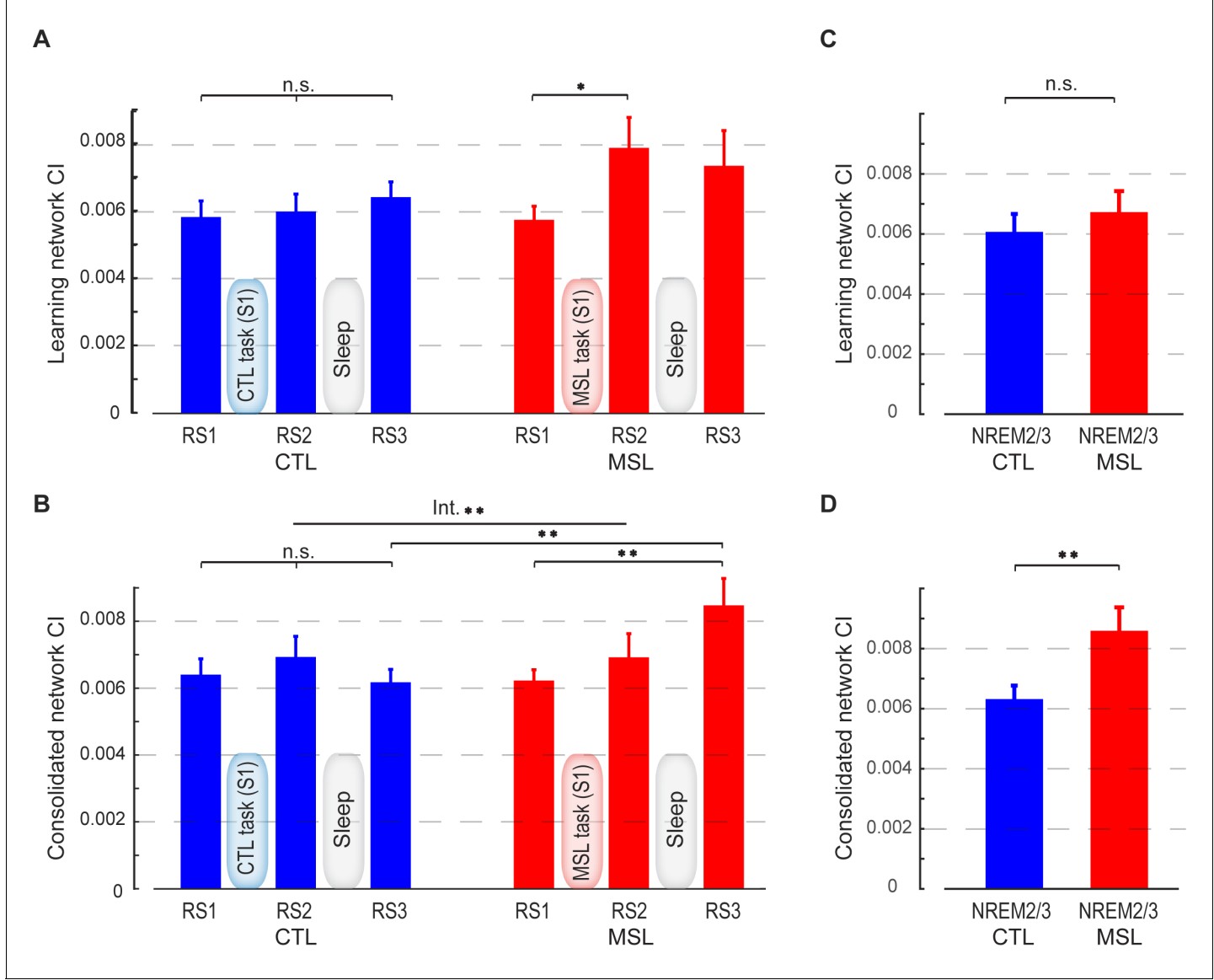

**Figure 2.** Connectivity index (CI) during resting-state and NREM sleep. (**A, B**) show the CI within the learning and the consolidated patterns, respectively, averaged across subjects during the resting-state conditions before (RS1) and after (RS2) the training session (S1), as well as on the following morning before retest (RS3). The learning pattern's CI was significantly increased in the MSL immediately following training (RS2), while the consolidated pattern's CI increased significantly only post-sleep (RS3) in the MSL as compared to the CTL condition. (**C, D**) show, respectively, CI within the learning and the consolidated patterns averaged across subjects during NREM sleep. Only the consolidated pattern's CI differed significantly between the MSL and CTL nights. Error bars represent s.e.m.; * and ** indicate $p<0.05$ and $p<0.01$, respectively.

The following figure supplements are available for figure 2:

**Figure supplement 1.** The group-level spatial maps of four highly-reproducible brain networks extracted during the MSL task.

**Figure supplement 2.** Connectivity index (CI) during resting-state periods within the four control networks reported in *Figure 2—figure supplement 1*.

**Figure supplement 3.** Connectivity index (CI) during non-REM (NREM) sleep within the four control networks reported in *Figure 2—figure supplement 1*.

networks as a function of motor task across resting-state periods (no interaction; p>0.25, and no main effect of task or resting-state condition; p>0.2, in all four networks; *Figure 2—figure supplement 2*). Likewise, no significant change in CI between the two tasks was found during NREM sleep in any of the four networks (paired *t*-statistics, p>0.3; *Figure 2—figure supplement 3*). Overall, these analyses suggest that the observed changes in the learning and consolidated patterns were not due to some global epiphenomena of time, learning or sleep on resting state connectivity.

## Neural substrates underlying memory reorganization during NREM sleep

The CI analyses allowed us to specify whether a pattern of brain areas as a whole showed changes in functional connectivity across different conditions. However, in order to specify the brain areas within the consolidated pattern that are primarily responsible for modulating the strength of connectivity in the MSL night, we performed a dual regression analysis (*Filippini et al., 2009*). For each subject, this approach projects a group-level activation map (i.e., as a spatial regressor) onto a selected fMRI condition (e.g., RS1 or RS3), in order to identify brain areas within that spatial map or pattern that are specifically recruited during the given fMRI run (see Materials and methods for more details). Separate dual regression analyses were carried out using the learning and the consolidated patterns as spatial regressors. For each pattern, a group-level repeated-measures GLM was performed to evaluate the contribution of different areas within each pattern during NREM sleep periods across tasks. We found that the ventrolateral putamen was the primary brain region within the consolidated pattern whose functional connectivity was significantly elevated during NREM sleep in the MSL compared to the CTL night (corrected for multiple comparisons using Gaussian random field theory (GRF), p<0.05, *Figure 3*). Importantly, changes in functional connectivity between the putamen and the rest of the structures in the consolidated pattern were significantly related to the amount of sleep-dependent behavioral gains in motor performance ($r = 0.72$, p = 0.005 (N = 13), *Figure 3* scatter plot). This indicates that individuals with greater increases in functional connectivity with the putamen had greater overnight improvements in performance. By contrast, none of the brain areas within the learning pattern showed a significant change in connectivity during NREM periods between the two tasks.

Likewise, we used the dual regression method to identify the brain areas within each of the learning and consolidated patterns whose connectivity changed during either of the post-learning resting-state conditions (i.e., RS2 or RS3) as compared to the baseline (i.e., RS1) across the two tasks (resting-state conditions × task, repeated-measure ANOVA). Consistent with the active role of the putamen found during NREM sleep, this analysis revealed that the ventrolateral putamen (*Figure 4a*) and lobules V-VI of the cerebellar cortex (*Figure 4b*) were the principal brain regions responsible for the elevated connectivity within the consolidated pattern during the post-sleep resting-state condition (significant $(RS3 - RS1) \times (MSL - CTL)$ interaction, corrected for multiple comparisons using GRF, p<0.05). *Figure 4—figure supplement 1* shows the average amounts of connectivity in each cluster for each session and task separately. As shown in the figure, the amplitude of connectivity was significantly elevated in RS3 compared to RS1 only in the MSL task (paired *t*-statistics, p<0.01 in MSL for both clusters, p>0.2 in CTL for both clusters). Yet among these two structures, only the putamen connectivity was significantly correlated with the amount of overnight gains in performance speed ($r = 0.69$, p<0.01 (N = 13), *Figure 4a*, right scatter plot). Again, no area within the learning pattern showed a significant change in connectivity during the post-sleep resting-state condition across tasks. However, when the learning pattern was examined during the pre-sleep resting-state condition immediately following training (i.e., RS2), connectivity was elevated in the posterior parietal lobule with respect to the baseline condition in the MSL compared to the CTL task (significant $(RS2 - RS1) \times (MSL - CTL)$ interaction, p<0.05; *Figure 4—figure supplement 2*). Consistent with the CI analyses, none of the areas within the consolidated pattern showed changes in connectivity in the pre-sleep resting-state condition compared to the baseline, hence suggesting again that changes within the consolidated pattern were initiated during sleep.

## Temporal dynamics of the motor memory trace during NREM sleep

Given that we first observed an elevation of connectivity within the consolidated pattern during NREM sleep following MSL, we further examined the pattern of dynamic changes in CI over the

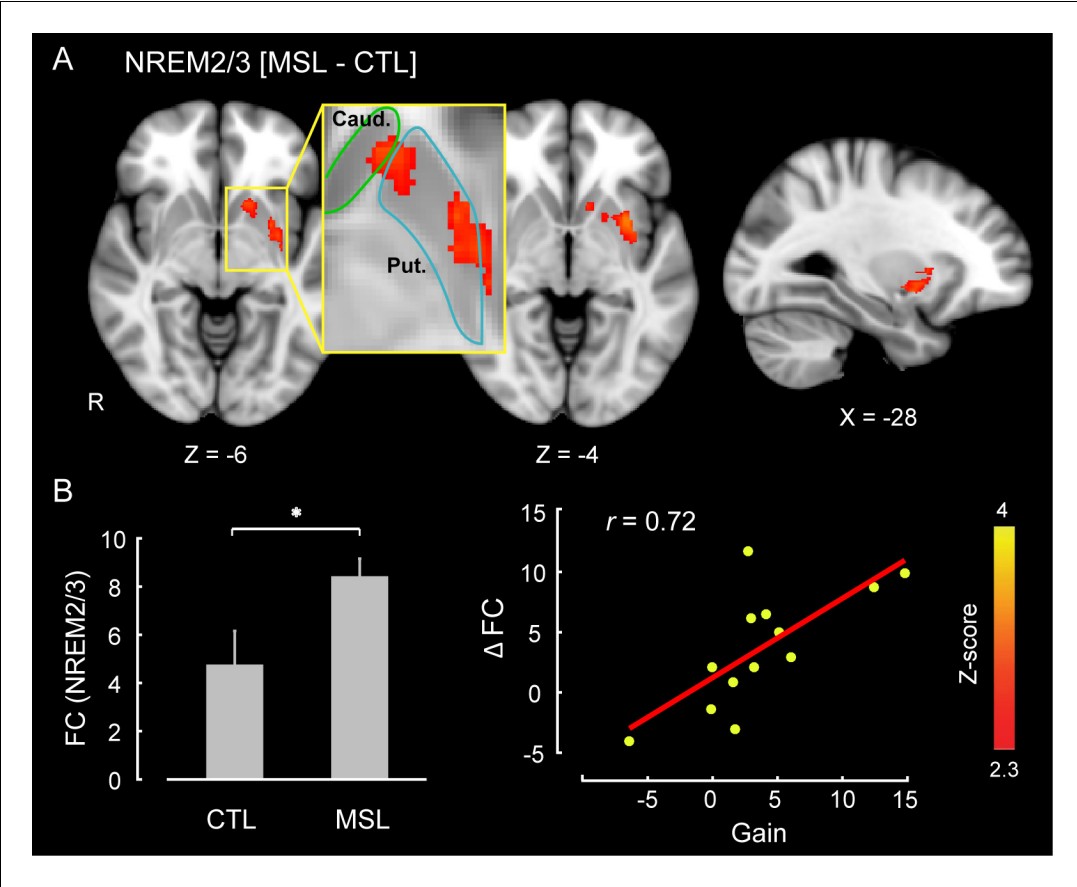

**Figure 3.** Neural correlates of motor sequence memory consolidation during NREM sleep. The ventrolateral putamen functional connectivity within the consolidated pattern differed significantly between the MSL and CTL nights during NREM sleep. Bar plot illustrates the functional connectivity of the putamen with the rest of structures in the consolidated pattern averaged across subjects during NREM sleep. Importantly, the putamen connectivity within the consolidated pattern during NREM sleep was significantly related to the extent of sleep-dependent performance speed gains on a per subject basis, as depicted in the scatter plot. The color-coded activation map indicates Z-score values and is corrected for multiple comparisons using GRF, p<0.05. Error bars represent s.e.m.; * indicates p<0.05.

The following source data and figure supplement are available for figure 3:

**Source data 1.** Regions of interest (ROI) used in the seed-based functional connectivity analysis.

**Figure supplement 1.** Changes in brain functional connectivity related to motor sequence learning during non-REM sleep.

course of the post-training night as compared to the intermittent periods of wakefulness. To do so, we employed a sliding-window approach over the temporally-concatenated fMRI data of each state and condition. Specifically, we separated NREM stage two and SWS periods, and calculated CI values in NREM stage 2, as well as, the intermittent periods (epochs) of wake, for which we had sufficient data in our group of participants. On average 22 min of data (600 volumes) for each subject and condition were selected (see Materials and methods for more details, and *Table 1* for sleep architecture information in the scanner). The mean epoch duration and the number of epochs selected for data analysis are reported in *Figure 5—source data 1*. Note that there was no significant difference in the characteristics of the selected epochs between MSL and CTL conditions (*Figure 5—source data 1*). Additionally, the mean epoch duration and the number of concatenated epochs were not significantly different between the wakefulness and NREM stage two sleep periods

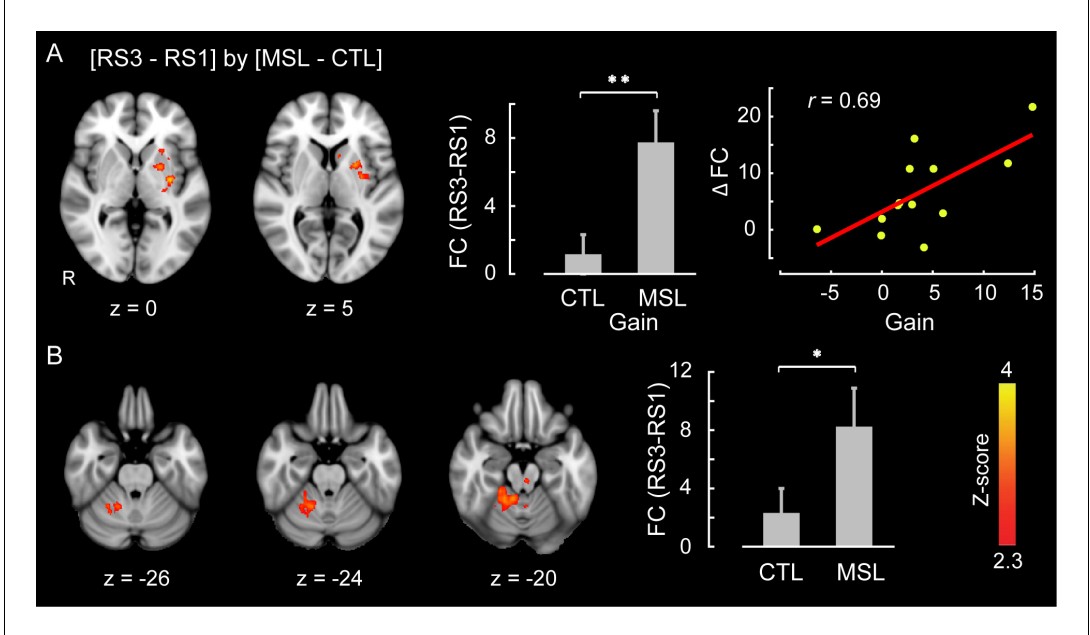

**Figure 4.** Neural correlates of motor sequence memory consolidation during post-sleep resting-state periods. The ventrolateral putamen (**A**) and the cerebellar cortex (lobules V-VI) (**B**) functional connectivity within the consolidated pattern differed significantly between the MSL and CTL conditions during post-sleep resting-state periods (RS3) as compared to baseline (RS1). Bar plot illustrates the change in functional connectivity of putamen within the consolidated pattern between RS3 and RS1 scans averaged across subjects in each task condition. The scatter plot in (**A**) shows that only the putamen functional connectivity was significantly related to the extent of overnight behavioral gains in performance speed. The color-coded activations maps indicate Z-score values and are corrected for multiple comparisons using GRF, p<0.05. Error bars represent s.e.m.; * and ** indicate p<0.05 and p<0.01, respectively.

The following figure supplements are available for figure 4:

**Figure supplement 1.** Functional connectivity within the consolidated pattern during post-sleep resting-state periods (RS3) and baseline (RS1) in each task.

**Figure supplement 2.** Neural correlates of motor sequence learning during resting-state periods immediately following training.

**Figure supplement 3.** Changes in brain functional connectivity related to motor sequence learning during the post-sleep (top row) and the pre-sleep (bottom row) resting-state conditions.

(p=0.12, paired t-test for the mean duration; and p=0.40, Wilcoxon signed rank test for the number

**Table 1.** Sleep architecture during post-training EEG-fMRI recording session on CTL and MSL condition nights. Mean and SEM values are reported in minutes. Sleep onset is calculated relative to the start of simultaneous EEG-fMRI recording. See also Materials and methods and Results for additional details. Slow wave sleep (SWS).

|  | CTL | | MSL | | MSL vs. CTL | |
|---|---|---|---|---|---|---|
|  | **Mean** | **SEM** | **Mean** | **SEM** | **T** | **P** |
| Wake | 53.2 | 8.98 | 66.9 | 7.02 | 1.68 | 0.12 |
| Stage 1 | 10.8 | 2.16 | 8.6 | 1.58 | 0.76 | 0.46 |
| Stage 2 | 42.7 | 5.85 | 34.1 | 5.13 | 1.75 | 0.11 |
| SWS | 17.0 | 4.88 | 10.2 | 3.77 | 0.95 | 0.36 |
| Sleep onset | 16.96 | 6.12 | 15.80 | 3.01 | 0.17 | 0.87 |

of epochs). These analyses revealed a gradual increase in the strength of connectivity within the consolidated pattern during stage 2 NREM sleep following MSL as compared to the CTL task, as supported by a repeated measure ANOVA (significant $\mathrm{time} \times \mathrm{task}$ interaction, $F_{10,120} = 1.98$, p<0.05; and paired-samples $t$ test comparing mean of the first three and the last three time points [corresponding to the first and the last 7 min and 12 s periods of recorded NREM stage-2 fMRI data, respectively], $t_{12} = 2.23$, p<0.05; **Figure 5b**). Two follow-up repeated measures ANOVAs during stage two sleep also revealed a significant main effect of time on the consolidated pattern's CI in the MSL ($F_{10,120} = 2.35$, p<0.05), but not in the CTL task ($F_{10,120} = 0.98$, p>0.4). Importantly, however, the results showed that the consolidated pattern's CI did not change during the intermittent periods of wake distributed throughout the sleep session (no $\mathrm{time} \times \mathrm{task}$ interaction, $F_{10,120} = 0.93$, p>0.5; no main effect of task or time, p>0.8 for both; **Figure 5e**).

Likewise, CI within the learning pattern changed as a function of motor task (MSL vs. CTL) over the period of NREM sleep (significant $\mathrm{time} \times \mathrm{task}$ interaction, $F_{10,120} = 2.00$, p<0.05; **Figure 5a**), but not during wakefulness ($F_{10,120} = 0.73$, p>0.6; **Figure 5d**). In contrast to the consolidated pattern, however, the CI within the learning pattern showed a significant difference between tasks only at

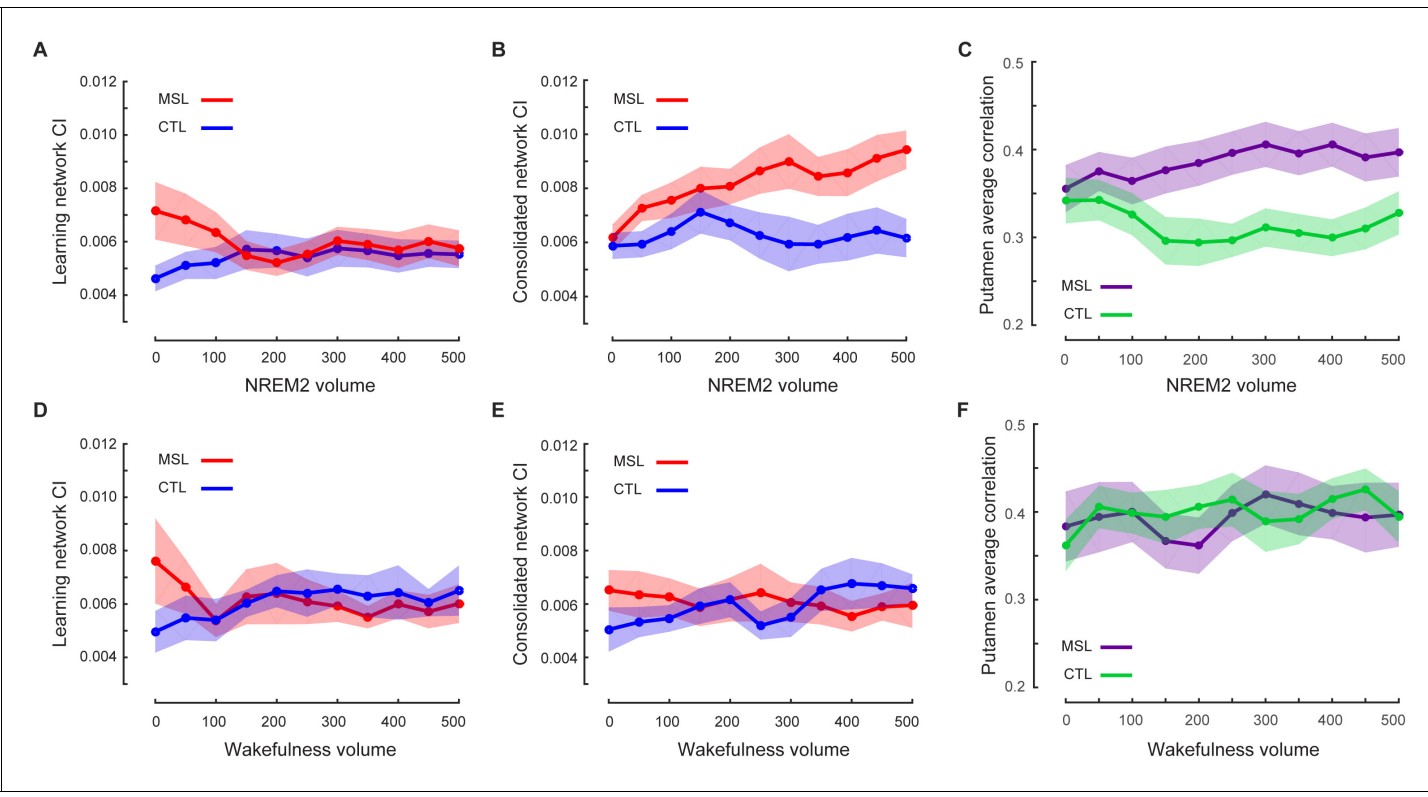

**Figure 5.** Temporal dynamics of memory trace during NREM stage two sleep. (A, B) illustrate the time course of CI change within the learning and the consolidated patterns during NREM stage two sleep, respectively. The consolidated pattern's CI gradually increased during NREM sleep only in the MSL condition (red curve), while the learning pattern's CI decreased. CI did not differ significantly over the course of NREM stage two sleep in the CTL night (blue curves). Furthermore, CI within the learning (D) and consolidated (E) patterns did not change significantly over the course of intermittent awakenings distributed throughout the sleep session in the CTL (blue curves) or MSL (red curves) night. (C), (F) illustrate the time course of ventrolateral putamen functional connectivity within the consolidated pattern during NREM stage two sleep and intermittent bouts of awakenings, respectively. Likewise, the putamen functional connectivity with the rest of structures in the consolidated pattern gradually increased only during NREM sleep in the MSL condition. Each data point is calculated using 100 fMRI volumes. Shaded area represents s.e.m.

The following source data and figure supplement are available for figure 5:

**Source data 1.** Average duration and number of epochs used in the temporal dynamics analysis (**Figure 5**).
**Figure supplement 1.** Robustness of temporal dynamics analysis with respect to the window size.

the beginning of the night. In fact, when early in the NREM stage two period was analyzed separately, we found a significant increase in connectivity within the learning pattern in the MSL compared to CTL (paired *t* test, mean of the first three time points (corresponding to the first 7 min and 12 s of NREM stage-2 fMRI data) in MSL compared to CTL, $t_{12} = 2.32$, p<0.05). However, there was no significant difference across tasks during later stage two sleep periods (mean of the last three time points in MSL compared to CTL; paired *t* test, $t_{12} = 0.47$, p>0.6).

In order to test the robustness of our findings with respect to the window size, we repeated our sliding window analyses iteratively by varying the size of the window. Consistently, a significant difference in the consolidated pattern connectivity was found between early and late periods in NREM sleep only following MSL using different window sizes ranging from 50 to 300 volumes (significant $\mathrm{windowsize} \times \mathrm{timeperiod(early vs. late)}$ interaction, $F_{5,60} = 5.67$, p<0.0005; and significant effect of time period (early vs. late) in the MSL, $F_{1,12} = 5.75$, p<0.05; *Figure 5—figure supplement 1*); thus suggesting that the pattern of results was robustly detectable over a range of time-windows.

In order to further examine the role of the ventrolateral putamen in shaping the functional connectivity of the consolidated pattern as supported by the dual regression analysis, we employed a similar sliding window approach to investigate the dynamic changes in functional connectivity of the putamen (same region as reported in *Figure 3*) within the consolidated pattern during NREM stage-2 sleep (see Materials and methods). We found that functional connectivity of the putamen was gradually enhanced during NREM two sleep following MSL as compared to the CTL task (significant $\mathrm{time} \times \mathrm{task}$ interaction, $F_{10,120} = 3.21$, $p = 0.001$; follow-up paired *t* tests comparing mean of the first three time points in MSL and CTL, $t_{12} = 0.89$, p>0.3; and mean of the last three time points in MSL compared to CTL, $t_{12} = 2.69$, p<0.05;*Figure 5c*), in line with CI analyses within the consolidated pattern. Again, there was no effect of task on the connectivity of putamen during the intermittent periods of wakefulness (no $\mathrm{time} \times \mathrm{task}$ interaction, $F_{10,110} = 0.59$, p>0.8; *Figure 5f*).

Finally, a series of conventional seed-based functional connectivity analyses (see Materials and methods) on the resting-state and NREM sleep data largely confirmed our findings using the dual regression approach (*Figure 3—figure supplement 1*, and *Figure 4—figure supplement 3*). The results revealed that increased connectivity of areas mainly within the consolidated pattern (including the putamen and cerebellar cortex; *Figure 1—figure supplement 2*, top row) during both NREM sleep (*Figure 3—figure supplement 1*) and the post-sleep resting-state condition ([RS3 – RS1]; *Figure 4—figure supplement 3*, top row) were significantly associated with the amount of overnight gains in performance. Also, the functional connectivity within the posterior parietal cortex (an area more activated during the learning compared to the retest; *Figure 1—figure supplement 2*, bottom row) was increased in the resting-state condition immediately following the MSL as compared to the baseline ([RS2 – RS1]; *Figure 4—figure supplement 3*, bottom row).

## Discussion

Several neurophysiological mechanisms have been proposed to account for the role of sleep in memory consolidation, including the trace reactivation, synaptic homeostasis, and systems consolidation hypotheses (see [*Frankland and Bontempi, 2005*; *Rasch and Born, 2007*; *Tononi and Cirelli, 2014*] for full review). Importantly, our findings are not only consistent with those processes, but they clearly demonstrate that consolidation of a motor sequence memory trace is mediated through interplay of these complementary mechanisms. First, our results point to a dynamic reactivation process, which began immediately after training during the subsequent resting-state period and extended into the early parts of NREM sleep. In line with these observations, several neuroimaging studies have documented elevation of functional connectivity in the post-training resting-state periods within the sensorimotor network implicated during motor learning (*Vahdat et al., 2011*; *Gregory et al., 2014*; *Sami et al., 2014*).

Likewise, several human neuroimaging studies during sleep have reported direct evidence demonstrating the reactivation of brain areas within the learning trace following visuospatial (*Rasch et al., 2007*; *Deuker et al., 2013*) and motor sequence learning (*Maquet et al., 2000*). However, there is increasingly more evidence supporting the view that memory trace reactivation is not specific to sleep, but also occurs in the post-training wakefulness before sleep (*Foster and Wilson, 2006*; *Peigneux et al., 2006*; *Karlsson and Frank, 2009*; *Carr et al., 2011*; *Staresina et al., 2013*; *Tambini and Davachi, 2013*; *Ambrose et al., 2016*). Our results extend these observations by

showing a general trace reactivation process involving the entire learning pattern during post-training resting-state periods, which returns back to baseline levels early in NREM sleep. These findings suggest that sleep enables two distinct complementary processes that take place in parallel, whereby sleep: (1) has a downscaling impact on the memory trace formed during learning at the systems level, and (2) actively transforms the trace to a consolidated state that is related to offline gains in performance. The downscaling of functional connectivity within the locally-activated learning pattern during the first hour of NREM sleep in our study is reminiscent of the results obtained using EEG recordings following a visuomotor adaptation task (*Huber et al., 2004*). Huber and colleagues showed that localized EEG slow wave activity within regions activated during motor adaptation temporarily enhanced, and then restored to a control night activity level during the first hours of NREM sleep. Likewise, our results are consistent with a network-level restoration process, that is, the elevated functional connectivity within a newly-acquired learning pattern is downscaled back to pre-learning baseline levels during NREM sleep, thus supporting the synaptic homeostasis hypothesis at a systems level.

Moreover, when we examined individual brain areas within the learning and consolidated patterns, we found that the ventrolateral putamen and lobule VI of the cerebellar cortex were mainly involved in the reorganization process following motor sequence learning. Critically, changes in functional connectivity of the putamen within the consolidated pattern during NREM sleep, as well as during the post-sleep resting-state periods, were related to the extent of overnight behavioral gains in performance. Similar to the consolidated pattern taken as a whole, the strength of functional connectivity in the putamen gradually increased over the course of NREM sleep following MSL. These findings highlight the central role of the putamen in the consolidation and reorganization of motor sequence memory during NREM sleep. These results are consistent with previous work investigating the neural substrates of motor sequence memory consolidation in both humans (*Doyon and Benali, 2005*; *Debas et al., 2010*) and animals (*Yin et al., 2009*; *Kawai et al., 2015*) models, which reported a shift in neural activity from cortical motor areas to subcortical structures, particularly, to the striatum. For instance, Debas et al. (*Debas et al., 2010*) observed a sleep-dependent enhancement in striatal activity during practice of a motor sequence from training to retest following sleep, but not wake. Our work complements these findings by showing that this reorganization process takes place gradually over the course of NREM sleep.

Another important observation was that only changes in functional connectivity within the consolidated pattern emerging during NREM sleep (and subsequently in the following morning) were related to the extent of overnight improvement in performance (*Figures 3* and *4a*, *Figure 3—figure supplement 1*, and *Figure 4—figure supplement 3* top row). By contrast, functional connectivity in the learning pattern immediately following training was unrelated to the subsequent off-line gains in performance (*Figure 4—figure supplement 2*, and *Figure 4—figure supplement 3* bottom row). This further suggests that the processes related to the reorganization of memory trace during NREM sleep specifically support off-line improvements in motor sequence performance. Also, in order to disentangle the effects of NREM stage2 from SWS, we concatenated different chunks of NRME2 sleep to obtain sufficient amount of data, and investigated changes in connectivity patterns during NREM2 sleep. The results of this analysis (*Figure 5*) confirmed our initial findings shown in *Figure 2*. Yet, we did not have enough SWS data to perform a similar analysis.

One limitation of the current study is that, due to the extensive scanning time required for each participant, we did not run a separate wake control group in which participants simply stayed awake between the learning and the retest sessions. This might limit the conclusions one can make regarding sleep versus wake transformations. Yet to address this shortcoming, we acquired sufficient and comparable amounts of data during both sleep and wake periods following learning in the sleep scanning session (*Table 1* and *Figure 5—source data 1*) and performed control analyses to investigate the specificity of our results in relation to NREM sleep as compared to the simple passage of time during wakefulness (*Figure 5*). In this analysis, the architecture (mean duration and number) of the concatenated epochs were similar between MSL and CTL conditions, as well as between the wakefulness and NREM stage two sleep (*Figure 5—source data 1*). Remarkably, our results showed that, first, the learning pattern's connectivity significantly decreased during NREM sleep, but not during wake periods, and second, the consolidated pattern's functional connectivity was specifically increased during NREM sleep, while no significant change was observed during wake periods. Overall, the results of these analyses suggest that NREM sleep, in contrast to simple passage of time, is

likely essential for active reorganization of the memory trace toward a consolidated representation. Furthermore, dual regression analysis results revealed a close association between the amount of connectivity changes within the consolidated pattern during NREM sleep and the amount of off-line gains in performance on a per subject basis (*Figure 3*).

It should be noted that as it was not possible to have subjects sleep in the scanner more than ~2.5 hr, the acquired EEG/fMRI data contained only NREM sleep in the first part of the night. Only a few studies have specifically looked at early vs. late sleep, and the results reveal that sleep in the latter part of the night is preferentially associated with memory consolidation (e.g., [*Plihal and Born, 1997*; *Walker et al., 2002*]). However, it has also been shown that early sleep is involved in procedural memory processing (*Gais et al., 2000*), and that early and late sleep may synergistically contribute to memory processing (*Stickgold et al., 2000*). Remarkably, even a very short amount of daytime sleep containing NREM sleep during a nap can afford the same benefit to memory consolidation as a whole night of sleep (e.g., [*Mednick et al., 2003*; *Korman et al., 2007*; *Nishida and Walker, 2007*; *Doyon et al., 2009*]). Importantly, in the present study participants were allowed to return to the sleep laboratory for the remainder of the night, and, thus, post-sleep changes in behavior and their neural correlates were examined in the following morning. While we could not study directly the changes in neural activity in the latter part of the night, our findings suggest that early sleep has an important role in memory consolidation.

The observed elevated functional connectivity within the consolidated pattern during wakefulness resting before the retrieval test (RS3) could be explained in two ways. First, a reactivation of the task-related network might be due to the subjective expectancy to perform on the task later, as this information was provided to the subjects prior to the start of the morning scanning session. Alternatively, it might be due to the process of consolidation of an encoded memory trace following sleep. The observation that the elevation of functional connectivity was correlated with offline gains in performance on a per subject basis (*Figure 4*), however, support the latter hypothesis. Future experiments are needed to fully address this question, when participants are scanned in the morning with no expectancy of subsequent retrieval tests.

Finally, although NREM sleep has been directly implicated in declarative and visuospatial memory consolidation in humans (*Marshall et al., 2006*; *Rasch et al., 2007*), the role of NREM sleep in procedural memory consolidation is still controversial (*Tucker et al., 2006*; *Rasch et al., 2007*), as suggested for example by the dual process hypothesis of sleep (*Maquet, 2001*). Recently, however, there is increasing evidence indicating the essential role of NREM sleep in motor skill memory consolidation (*Rasch et al., 2009*; *Cousins et al., 2014*; *Schönauer et al., 2014*; *Laventure et al., 2016*). Several studies have emphasized the role of sleep spindles (*Fogel and Smith, 2006*; *Ramanathan et al., 2015*) and other features of NREM sleep such as slow wave activity (*Cousins et al., 2014*; *Gulati et al., 2014*) in the consolidation of a procedural memory. Indeed, learning-dependent changes in sleep spindles take place following motor learning (*Fogel and Smith, 2006*) and are shown to be related to the amount of off-line gains in performance (*Nishida and Walker, 2007*), as well as enhanced activity in the putamen during NREM sleep (*Fogel et al., 2017*) and at retest following sleep, but not wake (*Fogel et al., 2014*). Altogether, this suggests that sleep spindle activity is a possible neurophysiological mechanism for driving the increased connectivity of the putamen within the consolidated pattern during NREM sleep. Yet it should be noted that due to the difficulties of having participants undergo a full sleep cycle in the scanner, it was not possible for us to record any EEG-fMRI data during REM sleep. Hence, we could not test the contribution of REM sleep to procedural memory in the current study. Future studies examining spindle-related neural activity, as well as, REM sleep following motor sequence learning are needed to directly address these questions.

## Conclusion

Our findings demonstrate a gradual shift in motor memory representations following motor sequence learning; a transiently activated cortical trace is downscaled back to baseline levels and a subcortically-dominant and more interconnected trace, emerges during NREM sleep. These findings suggest that sleep supports both a homeostatic restoration of the memory trace potentiated during learning, and also actively reorganizes the memory trace at a systems-level. Specifically, our findings reveal that the ventrolateral putamen plays a central role in the emergence of the consolidated pattern during NREM sleep.

# Materials and methods

## Participants

Thirteen healthy right-handed adults (seven female, age 27.4 ± 3.6; mean ± std) passed the inclusion/exclusion criteria (see below) and completed the full experimental protocol. Ethical and scientific approval was obtained from the Research Ethics Board at the Institut Universitaire de Gériatrie de Montréal (IUGM), Montreal, Quebec, Canada and informed written consent was obtained prior to entering the study. Subjects were included in the study based on the following inclusion/exclusion criteria. They had to be a non-smoker, medication free and to have normal body weight (BMI ≤25). They also had to present no history of psychiatric or neurologic disorders and to score ≤8 on the Beck Depression (*Beck et al., 1974*) and Anxiety (*Beck et al., 1988*) Inventories. Participants who had previous formal training as a typist or musician and who were categorized as extreme morning or evening types (Horne Ostberg Morningness-Eveningness Scale [*Horne and Ostberg, 1976*]), worked at night, or had taken a trans-meridian trip ≤3 months prior to the experiment were also excluded from this study. Finally, subjects were included if they did not exhibit signs of excessive daytime sleepiness (≤9 on the Epworth Sleepiness Scale [*Johns, 1991*]) and if the quality of their sleep was normal as assessed by the Pittsburgh Sleep Quality Index questionnaire (*Buysse et al., 1989*). Participants were required to keep a regular sleep-wake cycle (bed-time between 10:00 PM – 1:00 AM, wake-time between 07:00 AM – 10:00 AM) and to abstain from consuming alcohol, caffeine or nicotine and from taking daytime naps throughout their participation in the study. Compliance to the schedule was assessed using both sleep diaries and wrist actigraphy (Actiwatch 2, Philips Respironics, Andover, MA, USA) worn on the non-dominant wrist.

Moreover, one week prior to the first experimental session (*Figure 1a*), each participant experienced a screening night beginning at 11:00 PM in the mock scanner located at the Functional Neuroimaging Unit, Montreal, Quebec, Canada. Participants were given a two-hour opportunity to sleep. The mock scanner noise (recorded from the scanner and presented at same approximate sound level) and lighting conditions (i.e., lights off) were similar to those of the experimental nights in the actual MRI scanner. EEG signal was recorded using the same MR-compatible electrode cap as that during the experimental nights. Following this two-hour sleep opportunity in the mock scanner, EEG electrodes were removed and subjects were permitted to sleep in the nearby sleep laboratory until 7:30AM. In order to be included in the study, a minimum of five minutes of consolidated NREM sleep (i.e., the minimum amount of data necessary for data analysis purposes) during the two-hour screening period was required. Finally, 13 subjects met these inclusion criteria and completed the study, and were thus included in the data analyses (see power analysis at the end of Materials and methods).

## Finger motor sequence learning task

Subjects were tested using a version of the motor sequence learning task (*Karni et al., 1995*), in which they were required to perform self-generated finger movements with their non-dominant (left) hand as quickly and accurately as possible. A custom MR-compatible ergonomic response pad comprising four push buttons located in a row was used. Each participant was scanned under two different conditions including motor sequence learning (MSL) and control (CTL), which were performed on two separate nights and the following mornings one week apart (*Figure 1A*). The order of the MSL and the CTL conditions was counterbalanced across participants. In each condition, subjects first practiced the corresponding task in the evening (learning session, S1), and were tested again later on the same task in the following morning (retest session, S2).

On the MSL night, subjects first explicitly memorized and slowly demonstrated to the experimenter the 5-item sequence of finger movements (4-1-3-2-4, where one stands for the index finger and four for the little finger), until they could produce 3 consecutive correct 5-item sequences using an MR-compatible response pad. During the experiment, subjects lay supine in the scanner and executed the task following color-coded cues, which appeared on a screen visible via a mirror attached to the head coil. A green cross displayed in the center of the screen indicated the start of the task block, which terminated after 60 key presses. Each practice block was separated by a 15 s rest period (indicated by a red cross) during which subjects were instructed to keep their fingers immobile. Subjects were administered 14 blocks of practice during each of the learning and retest

sessions. All subjects performed the sequence with an average accuracy of more than 83%, corresponding to more than 10/12 correct sequences per block. The timing of all key presses was recorded and speed was measured by the inter-key-press interval for correct responses only.

On the CTL night, subjects were required to press all four fingers of the left hand simultaneously at the same average rhythm as the MSL task. This task was designed to have the same motor performance characteristics of the MSL condition (e.g., same number of finger flexion movements, same average inter-key press interval, all in the same amount of time), but importantly, without any sequence to learn. Similar to a previous study (*Orban et al., 2010*), random individual key presses were not used as we intended to employ a control task that was uncued and thus internally generated and explicitly known, similar to the MSL task. The use of self-generated random sequences was not possible either, as it has been shown that people are not able to reliably produce random sequences of movements (*Figurska et al., 2008*). Subjects were first instructed to press all four keys simultaneously following the rhythm of an auditory tone (presented monotonically at 3 Hz) as long as a green cross was displayed on the screen. This first pre-training step was intended to entrain subjects to the average speed of performance (~3 Hz) normally observed during practice of the MSL task. After three blocks of practice (60 movements each) in this pre-training step, subjects performed the task in the absence of the audio tones. Here, participants were instructed to maintain the same rhythm as practiced in the first pre-training step. Once performance was maintained at 3 Hz (±0.25 Hz) for three consecutive blocks, this step of the pre-training phase was terminated. This pre-training phase ensured that subjects could reliably press all four keys simultaneously at the target rhythm. Similar to the MSL task, the pre-training was not included in subsequent analyses. For the practice sessions of the CTL task, participants were instructed to follow the same rhythm as practiced during the pre-training phase, and to rest during the presentation of the red cross. Similar to the MSL task, subjects were administered 14 blocks of practice, where each practice block terminated after 60 simultaneous 4-key presses, and each intervening rest period lasted 15 s. Again to be consistent with the MSL task, performance in the CTL task was measured as the inter-response interval between consecutive key presses (i.e., simultaneous flexion of all four fingers). The onset of the first of four finger presses was used in the subsequent analyses if the four fingers did not precisely touch their respective keys instantaneously.

## Imaging parameters

Images were collected using a 3T TIM TRIO Siemens scanner with a 12-channel head coil. A structural volume was acquired in the sagittal plane using a magnetization prepared rapid gradient echo (MPRAGE) sequence (TR = 2300 ms, TE = 2.98 ms, FA = 9°, 176 slices, FoV = 256 $\times$ 256 mm$^2$, voxel size = 1 $\times$ 1 $\times$ 1 mm$^3$). For functional acquisitions, an echo-planar imaging (EPI) gradient echo sequence was used with the following parameters: TR = 2160 ms; TE = 30 ms; FA = 90°; FoV = 220 $\times$ 220 mm$^2$; matrix size = 64 $\times$ 64; 40 transverse slices, slice thickness = 3 mm; 10% inter-slice gap; inplane resolution = 3.44 $\times$ 3.44 mm$^2$. In order to minimize the effects of gradient artifact on electroencephalography recordings, the sequence parameters were chosen so that the MR scan repetition time (2160 ms) matched a common multiple of the EEG sample time (0.2 ms), the product of the scanner clock precision (0.1 µs) and the number of slices (40 slices). Imaging parameters were the same during the resting-state scanning periods (RS1 to RS4; *Figure 1A*), the practice sessions of the MSL and CTL tasks (S1 and S2), as well as post-training sleep where EEG measurements were simultaneously recorded with fMRI acquisitions. The number of acquired functional volumes during practice sessions was variable depending on the participant's speed during the task. Each resting-state scan, however, lasted for 150 volumes or 6 min and 24 s. The sleep session was terminated when the maximum possible number of volumes for a single fMRI session (4000 volumes, lasting a maximum of 2.5 hr) in the Siemens 3.0T TIM TRIO MRI system was reached, or if subjects voluntarily terminated the session. Similar to the acclimatization night, EEG electrodes were removed after this sleep opportunity and subjects were then allowed to sleep in the nearby sleep laboratory. The retest sessions were administrated at least 1.5 hr after awakening at 7:30 AM to ensure the dissipation of sleep inertia.

## Image preprocessing

Preprocessing of the imaging data was carried out using the FSL software package (*Beckmann et al., 2003*) and in-house programs developed in MATLAB. This included (1) removal of the first two volumes in each scan series, (2) slice time correction, (3) non-brain tissue removal (4) motion correction, (5) global intensity normalization, (6) spatial smoothing (Gaussian kernel of FWHM 6 mm) and (7) temporal high-pass filtering (σ = 100 s). To achieve the transformation between the low-resolution functional data and the standard stereotaxic space (MNI152: average T1 brain image constructed from 152 normal subjects), we performed two transformations. The first was from the functional image to the T1-weighted structural image (using a 6 degree of freedom (DOF) transformation), and the second was from T1-weighted structural image to the average standard space (using a 12 DOF linear affine transformation, voxel size = 2 × 2 × 2 mm). Also to minimize the effect of head motion during the sleep session data acquisition, those fMRI volumes that were scored as movement artefact in the EEG scoring (less than 1% of total volumes in all subjects, see Polysomnographic Recording and Analysis) were not included in the fMRI analyses.

In our functional connectivity analysis on sleep data during NREM periods as classified by the EEG scoring (see the EEG analysis for more details) and intermittent wakefulness periods in between sleep stages, we selected the longest continuous segment of data to avoid discontinuities/sudden jumps in our data analysis. Also, because we performed a repeated measures analysis, we selected for each subject the minimum number of sleep or wakefulness volumes available in each task night (MSL and CTL), so that the differences in the number of volumes across tasks did not affect our between-task contrast. These criteria resulted in the selection of 316 ± 32 (mean ± s.e.m.) fMRI volumes during NREM stage two and SWS, and 358 ± 17 (mean ± s.e.m.) volumes during wakefulness periods.

## Functional task sessions

For each subject, each condition (MSL or CTL), and each practice session (S1 or S2), changes in brain regional responses were estimated using a model including responses to the task practice blocks weighted by each block's performance speed (inversely related to block duration). This regressor was then convolved with a double-gamma hemodynamic response function (HRF). Six rotation and translation motion parameters were also included in the model as confounds. Because we were interested to extract a single map for each practice session, we did not separate areas related to the main effect of practice and those related to the modulation by performance speed. Hence, the use of a weighted regressor made this analysis more sensitive to identify all brain regions that are related to both task execution and performance improvements over the course of training. The subject-level regression coefficients (COPEs in FSL) and their variance maps (VARCOPEs in FSL) were then input to a group-level analysis, which used a mixed-effects (FLAME1, FSL) general linear model (Z > 3.5, corrected family-wise error using Gaussian random field theory, cluster significance threshold of p<0.05). In order to extract group-level maps of learning (S1) and retest (S2) practice sessions related to motor sequence learning as compared to simple motor performance, linear contrasts were performed to compare the difference between tasks during learning S1[MSL - CTL] and retest S2[MSL - CTL]. In this repeated-measures analysis, several regressors modeled the mean of each subject across different practice sessions. We name these thresholded Z-statistics maps (Z > 3.5) corresponding to S1 and S2 sessions the 'learning' and the 'consolidated' patterns, respectively. In each of the learning and consolidated patterns, we calculated the total volume of activated voxels, the volume of cortical activation using a mask extracted from the Harvard-Oxford cortical structural atlas (*Desikan et al., 2006*) (more than 25% tissue probability), and the volume of subcortical activation using a mask extracted from the Harvard-Oxford subcortical structural atlas and the probabilistic cerebellar atlas (*Diedrichsen et al., 2009*) (more than 25% tissue probability).

## Within-network functional connectivity index

To assess the strength of functional connectivity within a network of brain regions, we used a 'connectivity index' (or CI) as proposed in Vahdat et al. (*Vahdat et al., 2014*). Assume that $X(v)$ specifies a vector of voxel intensities for a given brain network including $n$ voxels, $v_i$ , $i = 1, .., n$, and $Y(v, t)$ represents a matrix of preprocessed BOLD data for a given subject and run (e.g. during resting-state or sleep condition) including $m$ volumes, $t = 1, .., m$. Data preprocessing included the regular steps

explained above, as well as normalization of each voxel time series by the standard deviation of all the voxels' time series inside the brain mask for each fMRI run and each subject. In this way, we made sure that between-run comparisons were not confounded by differences in total variation of BOLD signal across all brain voxels and time. We then first employed a spatial general linear model (GLM) to estimate the time course of activation of $X(v)$ in the BOLD data $Y(v,t)$ over time (**Equation 1 a**), and then normalized the resulting regression coefficients $\beta(t)$ using the standard deviation of the residuals $\varepsilon(v,t)$ (**Equation 1 b,c**). The normalized factor $\eta(t)$ follows a student-$t$ distribution, so that we could compare it over time and across different runs for a given network (**Kruggel et al., 2002**).

$$
\begin{aligned}
Y(v,t) &= X(v)\beta(t) + \epsilon(v,t) &&(a)\\
\sigma(t) &= std(\varepsilon(v,t)) &&(b)\\
\eta(t) &= \frac{\beta(t)}{\sigma(t)} &&(c)
\end{aligned}
\tag{1}
$$

Finally, the CI was calculated as the power (variance) of the resulting normalized coefficients $\eta(t)$ for each subject and each run. This index simply represents the strength of functional connectivity (or functional integration) of spatial pattern $X(v)$ in a given fMRI data $Y(v,t)$. The normalization by the regression residuals makes this index specific to the co-activation of the desired pattern ($X(v)$), and not to any general activation in all or parts of brain areas.

This connectivity measure was selected as opposed to other methods such as ICA and seed-based analysis due to the following reasons. First, unlike the application of data driven approaches such as ICA, we sought to examine specific hypotheses regarding the reactivation and reorganization of pre-defined activation patterns recruited during the learning and retest task performances. Second, we sought to investigate the reactivation of the entire learning/consolidated pattern using a multivariate approach, as opposed to univariate methods such as seed-based analysis, which are suited to examine distinct connections at each time. Third, although the representational similarity analysis (**Kriegeskorte et al., 2008**) offers a unique approach to study reactivation (**Staresina et al., 2013**; **Tambini and Davachi, 2013**), this approach is well-suited in designs aiming at studying activity patterns within a specific region of interest in the brain. As in this study we sought to investigate the reactivation patterns across all brain areas, a more general connectivity measure was desirable. Lastly, we sought to examine the dynamics of change in each specified network over time during sleep; hence estimation of a connectivity measure time-series was applicable. Nevertheless, we also report the results of seed-based functional connectivity analysis for comparison purposes.

We performed two separate ANOVA to compare changes in CI within each of the learning and the consolidated patterns. One analysis compared CI during NREM sleep between tasks (MSL – CTL), and the other one compared it across different resting-state conditions (RS1, RS2, and RS3) within each pattern. As the aim of the current study was to capture off-line changes in connectivity (whether related to simple passage of time or to sleep) that happens following the acquisition of a new motor sequence, thus, we did not include the final resting-state condition following retest session (RS4) in our analyses because it involved changes related to the retest practice session, which was beyond the scope and hypotheses of this study.

Furthermore, in a control analysis, we investigated changes in CI with respect to several highly-reproducible brain networks (**Damoiseaux et al., 2006**) during resting-state periods and NREM sleep. In this analysis, the preprocessed fMRI data registered to the MNI standard space during the MSL condition in both S1 and S2 sessions were temporally-concatenated across all subjects. This time-concatenated matrix was then fed to the fast-ICA algorithm (**Hyvärinen, 1999**) to extract 30 group-level spatial components. These Z-score spatial maps were correlated with the templates of four highly-reported resting-state networks including the default mode, visual, and left and right fronto-parietal networks (**Damoiseaux et al., 2006**), and the corresponding group-level spatial maps with the highest correlation with templates were selected (**Figure 2—figure supplement 1**). Similar to the learning and consolidated patterns, these group-level spatial maps were then used as the pattern of interest within which CI was calculated during each of the resting-state and NREM sleep conditions in both MSL and CTL nights. Similar ANOVA models as explained above were conducted to investigate changes in CI in each spatial map during resting-state periods and NREM sleep.

### Dual regression analysis

In order to identify the brain areas within a given network whose functional connectivity is significantly changed across different experimental conditions, we used dual regression analysis. The dual regression method is based on two levels of regression, the first level is similar to the spatial regression carried above (*Equation 1 a*), where for each subject and run, a time series of regression coefficients is estimated from a group-level spatial map (the given brain pattern). This time series is then normalized by its standard deviation, and is entered in a second level temporal regression as a predictor, where a spatial map is estimated for the same subject and run. These subject-level regression coefficients (COPEs in FSL) and their variance maps (VARCOPEs in FSL) were then input to a mixed-effects group-level analysis (FLAME1, FSL), where linear contrasts assessed the difference between task conditions (corrected for family-wise error using Gaussian random field theory, cluster significance threshold of $p < 0.05$). Again, we performed two separate group-level GLMs for each of the learning and the consolidated patterns. One analysis compared functional connectivity within each pattern during NREM sleep between tasks (MSL – CTL), while the other one used a repeated-measures ANOVA to assess changes in functional connectivity across different resting-state conditions (RS1, RS2, and RS3).

### Temporal dynamics connectivity analysis

To evaluate changes in the strength of functional connectivity in a given brain pattern over time, we calculated CI in a sliding window over the sleep fMRI session. In this analysis, for each subject, we selected NREM stage two sleep segments (epochs) of more than 50 volumes and concatenated all epochs in each run (note that we did not have enough data during SWS to perform this analysis). *Table 1* reports sleep architecture information inside the MRI scanner, including the average time spent awake and asleep, and in specific sleep stages, and sleep onset time relative to the start of simultaneous EEG-fMRI recording. We also applied the same concatenation procedure to the intermittent wakefulness periods (epochs) during the sleep run. This resulted in $954 \pm 156$ (mean ± s.e. m.) fMRI volumes during stage 2 of NREM sleep, and $1517 \pm 235$ (mean ± s.e.m.) volumes during wakefulness periods for each subject and each condition. We selected the first 600 volumes in each condition, so that we had enough and comparable number of fMRI volumes across subjects during stage two sleep or wakeful periods for group-level averaging (the mean duration and the number of selected epochs averaged across subjects for each of wakefulness and NREM stage two sleep in MSL and CTL nights are reported in *Figure 5—source data 1*). Then, we selected a window size of 100 volumes, which corresponded to 3 min and 36 s of data, and slid the window by 50-volumes steps (half the window size overlap), which resulted in 11 data points in each of the NREM stage two and wakefulness periods. We then calculated CI in each time window for the learning and the consolidated patterns. Finally, we performed two-factor repeated measures ANOVA (time by task) to assess changes in CI over time and experimental tasks (MSL and CTL). Similarly, in a region of interest (ROI) based analysis, we applied a similar sliding window approach to measure changes in the functional connectivity of a given ROI and the rest of consolidated pattern over time during NREM stage two sleep and wakefulness. Hence, in each time window, we calculated the Pearson's correlation between the mean time series of the ROI and each voxel inside the mask of the consolidated pattern (excluding the ROI voxels), and averaged the correlation values across all voxels. The average correlation values were then tested in a time (11 points) by task (two levels) repeated measures ANOVA.

### Seed-based functional connectivity

To ensure that the method employed here was reproducible using more conventional seed-based approaches (*Vahdat et al., 2011*, *2014*), we defined 12 ROIs based on peaks of activity during either the learning (6 ROIs) or retest (6 ROIs) practice sessions (*Figure 3—source data 1*). We defined a spherical mask (radius = 6 mm) around each seed in standard space. We re-sampled this mask first to the T1 weighted structural image of each subject and from there to the low-resolution functional space of that subject. For each subject, the average time course of the BOLD signal within the transformed mask during the resting-state (RS1, RS2, and RS3) and NREM sleep (stage two and SWS) was calculated.

The mean BOLD time-course of each ROI was used as a predictor in a subject-level GLM to assess the functional connectivity of that ROI with every other voxel in the brain. Physiological noise was removed from the fMRI data based on a procedure described in Vahdat et al. (*Vahdat et al., 2011*). We calculated the following regressors: the average white-matter BOLD signal (WM), cerebro-spinal fluid (CSF), and the global signal. In total, nine nuisance regressors were used: WM, CSF, global signal and six motion parameters (x, y, and z translations and rotations obtained from the motion correction step in preprocessing). Hence, for each subject and each run a separate multiple regression analysis was carried out using the time series of nuisance signals as confound regressors and the time series of the ROI as the regressor of interest. We included the time derivative of each ROI's signal as a regressor in the GLM to account for possible time differences in the haemodynamic response function (HRF) of different cortical areas, as well as the latency for signal propagation from one cortical area to another (*Vahdat et al., 2011*). This analysis produced maps of all voxels that were positively or negatively correlated with an ROI's mean time-course. This was followed by between-subjects analyses that were carried out using a mixed-effects model (FLAME1) implemented in FSL (*Beckmann et al., 2003*). As in Vahdat et al. (*Vahdat et al., 2011*, *2014*), we used each subject's behavioral outcome (overnight improvements in performance) as a regressor to obtain a weighted average of the difference between conditions (MSL compared to CTL). Corrections for multiple comparisons at the cluster level were carried out using Gaussian random field theory (min Z > 2.7; cluster significance: p<0.05, corrected). To correct for multiple ROIs we identified as statistically significant those clusters that had a probability level of better than p=0.05/12 (12 = number of ROIs).

We then examined the correspondence between the behavioral regressor and the changes in functional connectivity (for the resting-state data the change between RS3 or RS2 and RS1, and for the sleep data the difference between the MSL and CTL nights). We constructed a vector for each connection between an ROI and target cluster whose elements were each subjects' change in functional connectivity ($\Delta$FC). This vector was correlated with a vector of associated overnight improvements in motor performance.

## Polysomnographic recording and analysis

EEG was recorded by using an MR-compatible EEG cap (Braincap MR, Easycap, Herrsching, Germany) with 64 ring-type electrodes and two MRcompatible 32-channel amplifiers (Brainamp MR plus, Brain Products GmbH, Gilching, Germany). EEG caps included 62 scalp electrodes referenced to FCz. Two bipolar ECG recordings were taken from V2-V5 and V3-V6 using an MR-compatible 16-channel bipolar amplifier (Brainamp ExG MR, Brain Products GmbH, Gilching, Germany). Electrode-skin impedance was reduced to <5 KOhm using high-chloride abrasive electrode paste (Abralyt 2000 HiCL; Easycap, Herrsching, Germany). In order to reduce movement-related EEG artifacts, subjects' heads were immobilized in the head-coil by surrounding the subject's head with foam cushions. EEG was digitized at 5000 samples per second with a 500-nV resolution. Data were analog filtered by a low pass filter at 250 Hz and a high pass filter at 0.0159 Hz. Data were transferred via fiber optic cables to a personal computer where Vision Recorder Software, Version 1.x (Brain Products, Gilching, Germany) was synchronized to the scanner clock. Sleep EEG was monitored online with Brain Products RecView software using online artifact correction.

EEG data were preprocessed by a low-pass filter (60 Hz), down-sampled to 250 samples/sec and re-referenced to averaged mastoids. Scanner artifacts were removed using the 'fMRI Artifact rejection and Sleep Scoring Toolbox (FASST)' for MATLAB (Mathworks, Natick, Massachusetts, USA [*Leclercq et al., 2011*]), using an adaptive average subtraction method. Ballistocardiographic artifacts were then removed using an algorithm based on a combination of artifact template subtraction and event-related independent component analysis (*Leclercq et al., 2009*) for artifacts time-locked to the R-peak of the QRS complex of the cardiac rhythm. Following gradient artifact and ballistocardiographic artifact correction, EEG recordings were sleep stage scored according to standard criteria (*Berry et al., 2012*) to identify periods of NREM sleep, free of any movement artifact, during which the EEG and fMRI data were analyzed.

## Statistical analyses

Results are shown as mean ± s.e.m. Based on our previous resting-state analysis results on neurologically healthy subjects (*Debas et al., 2010*; *Vahdat et al., 2011*), on average, a difference of 0.73 ± 0.56 (mean ± std) of baseline functional connectivity (Z score units) within the sensorimotor network have been detected following motor learning. Thirteen subjects would thus provide 90% power to detect changes across experimental conditions at a significance level of α = 0.05 (*Rigby and Vail, 1998*). Data were checked for normality and equality of variance across conditions. Unless otherwise indicated, statistical significance was determined using repeated measures two-tailed *t* tests (when comparing two conditions) or repeated measures ANOVAs (when comparing more than two conditions). Results were considered to be significant at p<0.05.

## Additional information

### Funding

| Funder | Author |
| --- | --- |
| Canadian Institutes of Health Research | Shahabeddin Vahdat Julien Doyon |

The funders had no role in study design, data collection and interpretation, or the decision to submit the work for publication.

### Author contributions

SV, Formal analysis, Validation, Visualization, Methodology, Writing—original draft; SF, Conceptualization, Data curation, Validation, Writing—review and editing; HB, Conceptualization, Supervision; JD, Conceptualization, Supervision, Funding acquisition, Project administration, Writing—review and editing

### Author ORCIDs

Shahabeddin Vahdat, http://orcid.org/0000-0002-0494-6974
Stuart Fogel, http://orcid.org/0000-0002-3227-5370
Julien Doyon, http://orcid.org/0000-0002-3788-4271

### Ethics

Human subjects: Ethical and scientific approval was obtained from the Research Ethics Board at the Institut Universitaire de Gériatrie de Montréal (IUGM), Montreal, Quebec, Canada and informed written consent was obtained prior to entering the study.

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
