## [Decision Letter]

Thank you for submitting your article "Network-wide reorganization of procedural memory during NREM sleep revealed by fMRI" for consideration by *eLife*. Your article has been reviewed by two peer reviewers, and the evaluation has been overseen by a Senior Editor (Richard Ivry) and a Reviewing Editor. The reviewers have opted to remain anonymous.

The Reviewing Editor has drafted this decision to help you prepare a revised submission. We have good news in that the reviewers are generally supportive of the work. In particular, they are supportive of the approach used to examine the consolidation of procedural memory over time in as sleep/EEG/fMRI paradigm. However, there are a few issues that were brought up that would need to be adequately addressed in a revision.

1) The first, and most critical, issue has to do with the control task. One reviewer points out that this task likely involves some learning as well and that there is a known network of brain areas that contribute to learning in the control condition, some of which overlap with areas involved in sequencing (striatum, cerebellum, motor cortex, prefrontal cortex) and possibly serving different functions for these tasks. Thus the control condition may prohibit seeing sequencing-related changes because they are also timing-related. The reviewers noted that they can see why it is also a good control. For e.g. it also involves hand movement. Please show further data on the control task by unpacking the bar graphs in Figure 4, ideally showing (RS1 CTL = RS3 CTL = RS1 MSL) < RS3 MSL. Furthermore, please directly compare learning activation patterns in the two tasks, showing overlapping and non-overlapping regions, so that the readers (and reviewers) can get a better feeling of the control task. Also, it was noted that "The authors make the case that the CI in the consolidated pattern was enhanced for the motor learning task only after sleep (RS3) – but numerically the CI at RS2 falls between RS1 and RS3 and the pairwise difference between RS3 and RS2 (post sleep vs. pre-sleep) is not significant." Please address this comment.

2) Without a waking control condition, the conclusions regarding waking vs. sleeping need to be tempered. Since the waking bouts are shorter than the sleep bout, one cannot draw firm conclusions about sleep versus wake transformations.

3) One point for the Discussion is to acknowledge that no REM data were acquired. This is understandable in light of the difficulties of having participants undergo a full sleep cycle in the scanner, but from a theoretical point it would be highly interesting to compare non-REM and REM sleep in their contributions to procedural consolidation. In this case, please add discussion of this to the manuscript.

It would also be helpful to break up the Introduction into a few paragraphs. As it stands now, the Introduction contains a single long two page paragraph making it difficult to read.

---

## [Author Response]

*[…] 1) The first, and most critical, issue has to do with the control task. One reviewer points out that this task likely involves some learning as well and that there is a known network of brain areas that contribute to learning in the control condition, some of which overlap with areas involved in sequencing (striatum, cerebellum, motor cortex, prefrontal cortex) and possibly serving different functions for these tasks. Thus the control condition may prohibit seeing sequencing-related changes because they are also timing-related. The reviewers noted that they can see why it is also a good control. For e.g. it also involves hand movement. Please show further data on the control task by unpacking the bar graphs in Figure 4, ideally showing (RS1 CTL = RS3 CTL = RS1 MSL) < RS3 MSL.*

We have unpacked the bar plots in Figure 4, as requested by the reviewers. We have now added the following figure as a new supplemental figure (Figure 4—figure supplement 1) to present these results. We have added text to describe this point in the Results section as well.

*Furthermore, please directly compare learning activation patterns in the two tasks, showing overlapping and non-overlapping regions, so that the readers (and reviewers) can get a better feeling of the control task.*

Figure 1—figure supplement 1 shows the requested information, which is added as a new supplemental figure in the Results section. As shown in this figure (and also reported as a contrast map in Figure 1, top row), the main differences between the MSL and CTL activation maps during the learning session are related to increased activations in bilateral putamen, premotor cortex, posterior parietal cortex and more extended activity in the cerebellar cortex.

*Also, it was noted that "The authors make the case that the CI in the consolidated pattern was enhanced for the motor learning task only after sleep (RS3) – but numerically the CI at RS2 falls between RS1 and RS3 and the pairwise difference between RS3 and RS2 (post sleep vs. pre-sleep) is not significant." Please address this comment.*

We thank the reviewers for the suggested analysis. As suggested by the reviewer, we tested the pairwise difference between RS3 and RS2 in the consolidated pattern. We actually observe a significant increase in the consolidated pattern CI from RS2 to RS3 session in the MSL condition (significant MSL [RS3-RS2], paired-t-test, t_12_ = 2.26, p < 0.05). In support of our conclusions, there is no statistically significant difference between RS2 and RS1 (t_12_ = 0.94, p > 0.35), but there is a significant increase in CI from RS1 to RS3 (t_12_ = 3.14, p < 0.01). The new information is now added to the Results subsection “Sleep-dependent reactivation and reorganization of the memory trace”.

*2) Without a waking control condition, the conclusions regarding waking vs. sleeping need to be tempered. Since the waking bouts are shorter than the sleep bout, one cannot draw firm conclusions about sleep versus wake transformations.*

We thank the reviewer for the suggested comment. As suggested, we have tempered our conclusion regarding waking vs. sleeping in the Discussion. We have also clarified and unpacked this pattern of results to provide our view on the results reported in the sleep vs. wake segments of the EEG-fMRI post-training session. The modified information is as follows:

“One limitation of the current study is that, due to the extensive scanning time required for each participant, we did not run a separate wake control group in which subjects simply stayed awake between the learning and the retest sessions. […] Furthermore, dual regression analysis results revealed a close association between the amount of connectivity changes within the consolidated pattern during NREM sleep and the amount of off-line gains in performance on a per subject basis (Figure 3).”

Also, please note that, as reported in [Supplementary-material SD4-data], the waking bouts recorded during the sleep session were not significantly different from the sleep bouts, as described below:

“Additionally, the mean epoch duration and the number of concatenated epochs were not significantly different between the wakefulness and NREM stage 2 sleep (p = 0.12, paired t-test for the mean duration; and p = 0.40, Wilcoxon signed rank test for the number of epochs).”

This information has been clarified in both the Results and Discussion sections.

*3) One point for the Discussion is to acknowledge that no REM data were acquired. This is understandable in light of the difficulties of having participants undergo a full sleep cycle in the scanner, but from a theoretical point it would be highly interesting to compare non-REM and REM sleep in their contributions to procedural consolidation. In this case, please add discussion of this to the manuscript.*

The reviewer and editor are right here as this is a limitation of most of the EEG-fMRI recordings studies performed during sleep, as it is very difficult to have subjects sleep in the scanner environment for a full sleep cycle. Hence reaching and recording REM sleep was not possible in the current study. We now have added text in the Discussion to clarify the fact that we have not tested the contribution of REM sleep to procedural memory in our study:

“Yet it should be noted that due to the difficulties of having participants undergo a full sleep cycle in the scanner, it was not possible for us to record any EEG-fMRI data during REM sleep. Hence, we could not test the contribution of REM sleep to procedural memory in the current study. Future studies examining spindle-related neural activity, as well as, REM sleep following motor sequence learning are needed to directly address these questions.”

*It would also be helpful to break up the Introduction into a few paragraphs. As it stands now, the Introduction contains a single long two page paragraph making it difficult to read.*

Thank you for the suggestion. Done.